# Effect of Adding Molybdenum on Microstructure, Hardness, and Corrosion Resistance of an AlCoCrFeNiMo_0.25_ High-Entropy Alloy

**DOI:** 10.3390/ma18194566

**Published:** 2025-09-30

**Authors:** Mariusz Walczak, Wojciech J. Nowak, Wojciech Okuniewski, Dariusz Chocyk

**Affiliations:** 1Department of Materials Engineering, Faculty of Mechanical Engineering, Lublin University of Technology, Nadbystrzycka 36D, 20-618 Lublin, Poland; 2Faculty of Mechanical Engineering and Aeronautics, Rzeszow University of Technology, Powstanców Warszawy 12, 35-959 Rzeszów, Poland; wjnowak@prz.edu.pl; 3Department of Applied Physics, Faculty of Mechanical Engineering, Lublin University of Technology, Nadbystrzycka 36D, 20-618 Lublin, Poland; d.chocyk@pollub.pl

**Keywords:** high-entropy alloys, Mo effect, corrosion resistance, nanoindentation

## Abstract

Recent literature reports have shown that individual HEAs, especially those of the AlCoCrFeNi composition system alloyed with appropriately selected elements, exhibit excellent mechanical properties and corrosion resistance, making them promising candidates for replacing conventional materials such as austenitic steels in corrosive environments. Therefore, in the present study, the high-entropy alloy AlCoCrFeNiMo_0.25_ was examined and compared with AISI 304L steel and the reference alloy AlCoCrFeNi. The HEA was produced by arc melting in vacuum. The effect of molybdenum addition (5% at.) on the structure, mechanical properties, and corrosion resistance was evaluated. Potentiodynamic polarization and electrochemical impedance spectroscopy tests were carried out in a 3.5% NaCl solution in a three-electrode electrochemical system. The addition of molybdenum to AlCoCrFeNiMo_x_ alloy additionally caused, along with the BCC phase, the formation of σ phase and FCC phase (less than 1%), as well as changes in the microstructure, leading to the fragmentation of grains and the formation of a mosaic structure. On the basis of nanoindentation tests, it was established that the addition of Mo increases hardness and elastic modulus and improves nanoindentation coefficients H/E and H^3^/E^2^, as well as an increase in the elastic recovery index while decreasing plasticity index (vs. the reference equiatomic HEA). This indicates the improvement of anti-wear properties with impact loading resistance. In turn, electrochemical tests have shown that the addition of Mo improves corrosion resistance. Corrosion pitting develops in Al- and Ni-rich areas of HEA alloys, as a result of galvanic microcorrosion related to Cr chemical segregation. In general, the addition of 5% Mo results in a fine-grained mosaic structure, which primarily translates into favorable nanoindentation and corrosion properties of the AlCoCrFeNiMo_0.25_ alloy.

## 1. Introduction

Degradation of materials under various operating conditions, especially in corrosive flow environments, can seriously affect the durability of the components [1]. During use, components are required to withstand various forms of degradation, such as cavitation, abrasive wear, and corrosion. In turn, flow machine components such as pipelines, valves, rotors, and turbines are severely exposed to various forms of deterioration, such as slurry erosion, cavitation, and degradation under corrosive conditions, which intensify their deterioration process [2]. The problems associated with various wear processes require high corrosion resistance from the materials in addition to favorable mechanical properties. Among engineers, stainless steels have so far been the first choice for applications in aggressive corrosive environments [3]. The recent interest in HEAs (high-entropy alloys) may provide an alternative for use in corrosion-resistant steels [4]. Although the corrosion resistance of stainless steels is at an acceptable level, they have relatively limited and sometimes insufficient mechanical properties [5]. Therefore, HEAs with properly selected alloying elements can outperform stainless steels in performance.

The literature describes experiments conducted on hundreds of different alloy compositions with high entropy [6,7]. Not all HEAs exhibit equally good mechanical and corrosion properties. AlCoCrFeNi was one of the first high-entropy alloys studied [8], and interest in this alloy composition is mainly due to its good mechanical properties [9,10]. The properties of this HEA composition can be shaped by changing the molar (or atomic) proportions of the constituent elements that make up the base alloy or by using new additives. The biggest influence on the structure and properties of the Al_x_CoCrFeNi alloy is primarily aluminum, where increasing its content in the alloy leads to an increase in strength while decreasing ductility [11]. The increase in aluminum content in the Al_x_CoCrFeNi alloy also leads to changes in the structure, i.e., the FCC (face-centered cubic) phase transitions to the BCC (body-centered cubic) phase [12]. Three variants of the structure are then possible, depending on the proportion of x (in molar ratio), i.e., when x ≤ 0.4 there is an FCC phase; a mixed FCC and BCC phase appears between 0.5 ≤ x ≤ 0.9; and when x ≥ 0.9 a single BCC phase is present [13]. HEA alloy strengthening is associated with network distortion due to the larger size of Al atoms compared to other major alloying elements [14,15].

HEA Al_x_CoCrFeNi alloys (like austenitic steels) are also expected to be corrosion resistant due to the content of elements that facilitate the formation of stable passive layers, mainly Cr [16]. As indicated by Shi et al. [17] all common corrosion-resistant alloys, including HEAs, can nevertheless undergo local pitting corrosion. Thus, according to Lin and Tsai [18], the proportion of 0.5 mole of Al in the Al_0.5_CoCrFeNi alloy leads to the formation of an Al-Ni rich phase (BCC), which is susceptible to galvanic corrosion in a 3.5% NaCl solution. Also, studies of Shi [17] in the alloys for the variants x = 0.5 and x = 0.7 confirmed that pitting is being generated in the BCC structure. In contrast, in the case of single-phase alloys with an FCC structure for x = 0.1 and x = 0.3, pitting rarely forms, and the alloys demonstrate extremely high pitting potential (*E_pit_*) and strong corrosion resistance. Also, corrosion tests conducted by Kao et al. [19] in sulfuric acids for options x = 0, 0.25, 0.5, and 1.0 (in molar ratio) revealed that higher Al content determines lower values of pitting potential.

A second solution related to shaping the properties of alloys based on Al-Co-Cr-Fe-Ni composition is the introduction of alloying additives such as Si (reduction in friction coefficient and wear rate) [20], Nb (increase in yield strength and hardness) [21], V (increase in wear resistance) [22], B (better magnetic behavior) [23], Zr (improved mechanical properties) [24], Cu (increase in compressive strength) [25], Ti (increase in hardness and erosion resistance) [26] Mn (improvement of corrosion resistance and solid-solution strengthening) [27], or Mo (improvement of mechanical properties) [28].

For conventional materials such as austenitic steels, the addition of Mo is believed to improve the stability and repassivation abilities of the passive films. This beneficial effect is mainly explained by the fact that Mo modifies the polarity of the passive film by forming of molybdates and preventing Cl^−^ ingress, rendering the passive film more stable against breakdown [3]. The research of Dai et al. [29] demonstrated that in the case of FeCoCrNiMo_x_ high-entropy alloys, the addition of Mo enhances the Cr_2_O_3_/Cr(OH)_3_ ratio and Mo oxides are incorporated into the passive film. In the case of Al_0.5_CoCrFeMo_x_Ni alloy characterized by a mixture of FCC and BCC phases, the addition of Mo enhances formation of the σ phase, which translates into an improvement in hardness and compressive strength of cast alloys and a reduction in ductility of alloys [30]. Unfortunately, too much Mo content is responsible for the formation of the σ phase, which can cause a reduction in corrosion resistance [30,31]. It has been observed that the significant presence of Mo can reduce corrosion resistance due to localized corrosion in areas depleted of Cr and Mo.

Shang et al. [31] pointed out that, in the case of corrosion resistance, it is extremely important to determine the range of Mo content for equiatomic alloys with high entropy, taking into account the best proportion of Mo and σ-phase. As for the available reports related to Mo_x_ alloying of equiatomic AlCoCrFeNi, only the work of Zhu et al. [28] and one study by the authors [5] are available. Zhu’s team studied the effect of Mo on the microstructure and strengthening of alloys. They proved that the addition of Mo increases the strength of the alloy while decreasing its ductility. In contrast, in our earlier paper [5], the characterization of AlCoCrFeNiMo_x_ alloy (where x = 0, 0.05, 0.1, 0.15, 0.2, 0.25 in molar ratio) in the context of applications in flow environments, including wear sliding and cavitation erosion with limited corrosion characteristics, was presented. Based on preliminary mechanical tests (Vickers hardness and sliding wear) and polarization tests in that study, the Mo0.25 variant showed the greatest application potential in terms of the results obtained. Therefore, it was decided to characterize it more extensively in terms of mechanical properties and corrosion resistance. It should be noted that the Mo0.25 composition is novel in terms of chemical composition and has not yet been studied by other scientists.

This article represents a new approach by the authors in the context of the available literature data, and the current preparation is a broader development of the characterization of AlCoCrFeNi alloy modified with 5% at. Mo, which provides new insights into the structural and mechanical properties based on nanoindentation tests, including corrosion resistance in environments rich in noxious Cl^−^ ion.

## 2. Materials and Methods

### 2.1. Materials

High-entropy AlCoCrFeNiMo_x_ alloys (where x = 0 and 0.25 in molar ratio) were prepared using high-purity (99.99% by weight) Al, Co, Cr, Fe, Ni, and Mo metals as raw materials. The first stage of the work was to weigh each alloy element using a scale with an accuracy of 0.1 mg, so as to obtain the chemical composition given in Table 1. The powders were then mixed in a ball mill for 2 h in an Ar atmosphere (5N purity) to obtain the uniform mixtures of metallic powders. The powder mixtures were cold-pressed in a hydraulic press, molding cylinders with a diameter of 15 mm and a height of about 15 mm. The next stage involved melting cylindrical moldings in an Arc-Melter Edmund Bühler GmbH arc furnace under an argon atmosphere, using a 5-fold remelting of each molding to improve the chemical homogeneity. From the castings prepared in this manner, specimens in the shape of disks with a diameter of 16 mm and a height of 4 mm were cut by wire EDM (ZAP BP 05d, Końskie, Poland). Due to the fact that HEA alloys are an alternative to stainless steel applications, AISI 304L stainless steel was additionally used as a reference material, with the measured chemical composition listed in Table 2.

### 2.2. Microstructure Analysis

The X-ray diffraction (XRD) pattern was used for phase identification, grain size determination, and stress analysis on the surface of the AlCoCrFeNi and AlCoCrFeNiMo_0.25_ samples. XRD measurements were performed in Bragg–Brentano geometry (θ–2θ), using a high-resolution diffractometer (Empyrean Panalytical, Almelo, The Netherlands) and recorded using CuK_α_ (l = 1.5418 Å) radiation with K-Beta Ni-filter. The data were collected over the range 2θ = 20 ÷ 100° with a step size of 2θ = 0.01° and a counting time of 6 s per data point, using a proportional detector. The fixed divergence slit of 1/4° was used together with a 5 mm beam mask, and all scans were carried out in a continuous mode. Incident and receiving Soller’s slits were 0.04 rad. The crystalline phase in the samples was identified using High Score Plus software v.3.0 package with Crystallography Open Database. XRD patterns of both samples were processed to remove its background noise for peak matching and determining the full-width half-maximum (FWHM) of the corresponding peaks. The Williamsons–Hall plot (W–H plot) method was adopted to calculate the crystallite size of the rich tetragonal σ-phase and its corresponding microstrain in each sample.

The microstructure analysis was performed using a Nova NanoSEM 450 scanning electron microscope (FEI, Eindhoven, The Netherlands) with an EDS detector for elemental composition analysis (EDAX). Observations were made with the GAD detector (gaseous analytical detector, FEI, Eindhoven, The Netherlands). In addition, surface chemical composition analysis (SEM-EDS mapping) was performed on untreated samples using a Phenom ProX scanning electron microscope (Phenom-World, Waltham, MA, USA) with EDS (energy-dispersive X-ray spectroscopy) detector.

### 2.3. Hardness and Nanoindentation Testing

The hardness was measured on the surface of alloys using a Vickers FM-800 microhardness tester provided with the automatic system ARS 900 (Future-Tech Corp., Kawasaki, Japan). The load used in hardness testing was 9.807 N (HV1) with a dwell time of 15 s. In total, 20 indentations were made for every batch of specimens and the average value of all the measurements was then calculated. The nanohardness of test alloys was measured by the Olivier–Pharr method (Q&P) [32]. Measurements were made in compliance with PN-EN ISO 14577-1 [33] using Anton Paar’s Ultra Nano Hardness Tester (UNHT). The surface of the test samples was examined with a Berkovich diamond tip. The load in the test was increased at a steady rate of 20 mN/min from the moment the indenter came into contact with the examined surface until a force *F*_max_ was reached. For the contact to take place, the following condition had to be met, i.e., the limit rigidity had to exceed 150 µN/µm. The force *F*_max_ was maintained for 10 s, and then the indenter was unloaded at a steady rate of 20 mN/min. The force *F* and the indenter penetration depth *h* were measured. At least 30 tests per every test sample were performed with a force *F*_max_ of 10 mN. In addition to the standard, most commonly determined mechanical quantities such as *H* (nanohardness), *E* (elastic modulus), index *H/E* (elastic strain to fracture), *H*^3^/*E*^2^ (yield pressure), the elastic recovery index (*ERI*), and the plasticity index (*PI*) were also calculated based on the total work performed by the indenter (Equation (1)). The amount of total work performed by the indenter during pressing can be expressed as in Equation (1) below:(1)$$W_{total}=W_{elast}+W_{plast}$$
where *W_elast_* and *W_plast_* represent elastic energy and plastic energy, respectively, determined from the field under the loading–unloading curve of nanoindentation. In turn, the *ERI* and *PI* indicators were determined from Equations (2) and (3):(2)$$ERI=\frac{W_{elast}}{W_{total}}$$(3)$$PI=\frac{W_{plast}}{W_{total}}$$

### 2.4. Corrosion Testing

Electrochemical measurements of the tested materials were carried out in a solution of 3.5% NaCl at ambient temperature (~25 °C). The electrochemical properties were tested in a three-electrode system consisting of: a sample as the working electrode, a Pt plate as the counter electrode and a saturated calomel electrode (SCE) as the reference electrode. The Atlas Solich ATLAS 0531 Electrochemical Unit & Impedance Analyzer station was used for electrochemical testing. Measurements of the open circuit potential (OCP) were conducted for 60 min to reach a steady-state potential for all samples.

Electrochemical impedance spectroscopy (EIS) experiments were carried out at OCP from 100 kHz to 100 mHz with an AC amplitude of 5 mV. The results of EIS were determined using AtlasLab software v.2.29 to fit the corresponding equivalent circuit. The polarization curves were recorded with an automatic potential shift of 1 mV/s in the range from −600 mV to +300 mV. The results of the electrochemical measurements were analyzed in AtlasCorr software v.1.04. The values of parameters such as corrosion current density *i*_corr_ and corrosion potential *E*_corr_ were estimated from Tafel curves by analyzing potentiodynamic curves in the AtlasLab program. The pitting potential *E*_pit_ was determined by noting the potential at which a continuous increase in anodic current occurred, indicating sustained localized breakdown of the passive film. The electrochemical measurements were repeated three times to ensure the reproducibility of the result.

On the basis of the current density value *i*_corr_, the corrosion rate *CR* was calculated. The calculation of *CR* was based on ASTM Standard G 102-89 [34], according to the formula:(4)$$CR=K\frac{i_{corr}}{\rho }EW$$
where *CR*—corrosion rate [mm·yr^−1^], *i*_corr_—current density of corrosion [μA·cm^−2^], *K*—Reaction rate constant = 3.27 × 10^−3^ [mm·g·μA^−1^·cm^−1^·yr^−1^], ρ—density in [g·cm^−3^], and *EW*—equivalent weight. In addition, the densities of AlCoCrFeN, AlCoCrFeNiMo_0.25_, and AISI 304L are 7.15 g·cm^−3^, 7.35 g·cm^−3^, and 7.90 g·cm^−3^, respectively.

The equivalent weight (*EW*) values were calculated from the formula:(5)$$EW=\frac{1}{\Sigma \;\frac{n_{i}\cdot f_{i}}{M_{i}}}$$
where

*n*_i_—valence of the i-th element in the alloy, *f*_i_—mass fraction of the i-th element in the alloy, and *M*_i_—the atomic weight of the i-th element in the alloy.

Calculations of the corrosion rate *EW* included only those components whose content in the alloy was not less than 1 wt.%. The calculated *EW* for AlCoCrFeN samples was 18.62, and for AlCoCrFeNiMo_0.25_ samples 18.98. Similarly, *EW* for AISI 304L is 25.30, which corresponds to the literature data [4].

## 3. Results and Discussion

### 3.1. Microstructure Characteristics

X-ray diffraction patterns of AlCoCrFeNi and AlCoCrFeNiMo_0.25_ high-entropy alloy samples are shown in Figure 1. XRD patterns of AlCoCrFeNi showed the diffraction peaks located at around 2θ = 31.25°, 44.67°, 64.91°, and 82.25°. As a result of analysis using the High Score Plus software v.3.0 package with Crystallography Open Database (COD), it was found that the peak positions in the XRD profiles correspond to phases B2/BCC (COD no. 9008802) with a cubic structure and the space group Pm-3m. The occurrence of a peak at 2θ = 31.25° and the absence of peaks around 2θ = 54.8° and 72.9° indicate incomplete phase ordering. However, XRD patterns of AlCoCrFeNiMo_0.25_ showed diffraction peaks confirming B2/BCC phases located at around 2θ = 31.25°, 44.67°, 64.91°, and 82.25°, and new relatively low-intensity peaks at 43.23° and 50.38° representing the FCC phase with a cubic structure and the space group F/m-3m (COD no. 7204808). Additionally, the XRD profile reveals peaks at 2θ = 42.16°, 44.61°, 45.67°, 47.93°, 49.29°, and 51.39°, representing the σ-phase with a tetragonal crystal structure and the space group P/42mnm (COD no. 2106167). The above results indicate that adding a small amount of Mo atoms significantly changes the structure of the AlCoCrFeNi high-entropy alloy. The individual phases are marked at the diffraction peaks in Figure 1.

Nair et al. [2] confirmed that the cellular regions of AlCoCrFeNi, which are enriched in Al–Ni and Al–Fe, show B2 ordering. Interestingly, they additionally observed a sigma phase for this alloy variant as well. In contrast, the occurrence of a σ phase in the equivalent AlCoCrFeNi alloy was not confirmed by XRD analysis by Zhu [28], who only found its occurrence with the addition of molybdenum in the amount of Mo0.2. Admittedly, Nair et al. [2] tested an HEA alloy in the form of a shell, while Zhu et al. [28] tested a casting. In contrast, our earlier work [5] did not identify peaks from the FCC phase (around 42.23° and 50.38°), presumably due to limited hardware capabilities from using the Miniflex II X-ray diffractometer made by Rigaku. However, XRD analysis performed with an Empyrean Panalytical high-resolution diffractometer showed the existence of new peaks in the XRD pattern representing the FCC phase, and in this study they are noticeable, although their contribution to the melt is at ~1% level.

The Williamson–Hall method allows us to determine the average grain size and microstrain based on broadening of diffraction peaks in the XRD patterns. This method assumes that the total broadening of the diffraction peaks is a sum of broadening caused by grain refinement and microstrain. The average grain size and macrostrain can be calculated using the Williamson–Hall (W–H) equation:(6)$$\beta \cos\left(\theta \right)=\frac{K\lambda }{D}+4\sin\left(\theta \right)\varepsilon$$
where *θ* is the diffraction angle, l is the wavelength of X-rays (l = 1.5418 Å), *K* is the Scherer constant, generally taken as 0.9, *D* is the grain size, and ε is the microstrain. *β* is the diffraction peak’s full-width at half-maximum (FWHM) in radians, corrected by the instrument FWHM. The instrument FWHM was determined using a standard Si reference sample. After measuring *β* of several diffraction peaks, a curve of *β* cos(θ) vs. 4 sin(*θ*) was plotted and fitted linearly by means of the least squares method. The slope of the straight line is the microstrain, and from the intercept of the straight line on the ordinate, the grain size was calculated. Detailed data on the average grain sizes and microstrain of AlCoCrFeNi and AlCoCrFeNiMo_0.25_ samples are summarized in Table 3.

The W–H analysis of the FWHM values of peaks did not show significant microstrain in any phase of the samples. The main influence on peak broadening is the grain size. The analysis shows that the average grain sizes for B2/BCC phases are smaller after Mo addition.

SEM microstructures are shown in Figure 2. The microstructure of the reference HEA samples (Figure 2a) contains both large and small grains, with some grains exhibiting an equiaxed shape. In addition, at higher magnification, it is observed that needle-like phases start to grow beside the grain boundaries, and certain needle-like grains also exist in the centers of the grains (Figure 2c). Similar nucleation of needles at the grain boundaries was demonstrated by Shi et al. [35] for AlFeCrNiMo_0.8_ coatings. EDS analysis of the HEA alloy microspheres (Table 4) showed that the grain boundaries are characterized by a higher concentration of Cr (point B), while the interiors of the grains are dominated by Al (point A). In the case of Mo samples, a structure with much finer grain size was observed (Figure 2b), where, at higher magnification, fragmentation within the grains arranged in a mosaic pattern was observed (Figure 2d). Zhu et al. [28] indicated a similar type of microstructure already observed at Mo0.1 content, describing it as a typical laminar eutectic structure. In addition, in the case of the AlFeCrNiMo_0.25_ alloy, light and dark areas of precipitates were observed in the microstructure. EDS analysis (Table 4) of these sites showed that the darker areas (point A) primarily have higher concentrations of Al and Ni. On the other hand, the brighter ones (point B) have higher proportions of Cr and Mo, with relatively lower concentrations of Al and Ni. As illustrated in Figure 2e, a structure identified as sigma phase separation is displayed. The chemical composition obtained for this phase in the EDS analysis (Table 4, spot C) is similar to the results obtained by Zhu et al. [28] for the sigma phase occurring in the Mo0.2 alloy variant. In contrast to the theoretical thermodynamic information determined using the ThermoCalc program presented in [36], in addition to the typical elements for this phase (i.e., Cr, Fe, Ni, and Mo), EDS analysis (which has its technical limitations), even at high magnifications, indicates elements from the matrix such as Co and small amounts of Al. In the present case, the interlayer distance of the phase was found to be in the range of 100–300 nm. It is important to note that Zhu et al. [28] observed that the interlayer spacing of this phase decrease with increasing Mo content in the alloy.

Figure 2f show the equiaxed coarse microstructure of AISI 304L stainless steel (after electrolytic etching), consisting of austenite with visible twins. The crystal structure of the austenite phase is FCC.

### 3.2. Mechanical Properties

Surface hardness measurements (Table 5) indicated an increase in average hardness values for all tested HEAs compared to AISI 304L steel. The addition of molybdenum resulted in a more than 3-fold increase in hardness compared to AISI 304L steel and an increase by half compared to the reference HEA alloy without Mo. The resulting increase in hardness for AlCoCrFeNiMo_0.25_ is related to solution hardening and grain size enhancement. Shi et al. [35] indicated that the element Mo, with its large atomic radius, added to the HEA composition causes a strong deformation of the crystal network and, consequently, a significant increase in the strain energy of the crystal network. The obtained Vickers hardness averages for AlCoCrFeNi correspond well with the results obtained by Nair et al. [2], who obtained a value of ~530 HV for this alloy.

The σ phase is unintentionally obtained. As its content increases, the proportion of the BCC phase decreases at the expense of the σ phase. Consequently, the total hardness of the HEA alloy increases, as the hardness of the σ phase is significantly higher than that of the BCC phase. As mentioned earlier, the σ phase is a very hard and brittle intermetallic phase. It increases the hardness of the alloy but reduces its ductility, as has already been demonstrated in studies [28,35].

In our previous studies [5], it was demonstrated that the hardness of the alloy also increases with Mo content, which entails a restructuring of the coarse-grained structure to obtain a fine mosaic-patterned structure. In addition to the presence of the sigma phase, it is primarily the formation of a plate-like structure that contributes to the increase in the phase boundary, which in turn translates into an increase in the strength of HEA alloys with Mo [5,28]. As illustrated (Figure 2 and Table 3), Mo is present in both light and dark plates, with an increased Mo content observed in the lighter plates. The sigma phase occurs in plates with an increased Mo content, i.e., light plates. As Zhu et al. [28] indicated, the strength of HEA material improves significantly with increasing Mo content, while the ductility of the alloy is weakened. This is due to the fact that the ductility of the sigma phase is lower than that of the BCC phase. Mo occurs in the disordered BCC phase. As a reminder, the atomic radius of Mo is 190 pm, and adding Mo as an element with a larger atomic size can increase the lattice constant of the disordered BCC phase. As a result, its value approaches the lattice constant of the ordered B2 phase. Furthermore, Zhu et al. [28] indicated that the increase in hardness and decrease in ductility of HEA alloys with Mo can be attributed to the fact that the number of slip systems in a complex structure (BCC + σ) is lower than that of a BCC structure. This also explains why the ductility of the σ phase is significantly lower than that of BCC.

Grain refinement in the case of the mosaic structure of the AlCoCrFeNiMo_0.25_ alloy may determine the Hall–Petch effect, contributing to strengthening of the material and increasing its strength [5]. As pointed out by Xu et al. [37], secondary phases influence strain hardening, the strength–ductility balance, and overall mechanical response. Xi and Li [38] indicated that the presence of the σ phase increases alloy strength by anchoring grain boundaries and impeding their movement. This, in turn, prevents the formation of twinning, which reduces ductility. The increase in material strength can also be explained by the effect of σ phase on grain boundaries, blocking both sliding and dislocation motion.

On the other hand, it is difficult to relate the hardness of the AlCoCrFeNiMo_0.25_ alloy to the literature data, since such an alloy composition has not been studied. Figure 3 illustrates the load–displacement curves, and Table 6 summarizes the results obtained from the nanoindentation tests. As can be observed from the curves, at 10 s intervals, all tested materials exhibit a tendency to creep, as the displacement of the diamond indenter increased at a constant maximum load of Fmax = 10 mN. Although creep changes are analyzed at much higher holding times [39], for HEA alloys based on the Al–Co–Cr–Fe–Ni composition, the authors of [40,41] used a holding time of 10 s to determine whether creep occurred. Jiao et al. [41] demonstrated that for an equimolar alloy, a significant accumulation of material around the indenter suggests that strongly localized plastic deformation occurs. In this type of HEA alloy, the deformation mechanism during nanoindentation is explained by an increase in dislocation density, which increases as the indentation depth decreases. The dislocations then spread to smaller slip circles, which can lead to nanoindentation creep.

On the basis of the load–displacement curves, it is apparent that stainless steel reached the highest maximum depth of penetration, whereas the AlCoCrFeNiMo_0.25_ alloy showed the lowerst. The smaller penetration depth of HEA materials (vs. AISI 304L) is an indication of lower plastic deformation.

For the AlCoCrFeNiMo_0.25_ alloy, a relatively high nanoindentation hardness with a relatively significant standard deviation is recorded (Table 6), which can be associated with the mosaic pattern of the structure and the presence of a σ phase. Molybdenum, which is regarded as a sigmoidal element, also results in higher Young’s modulus values. As demonstrated in a study by Zhuang et al. [30], although the formation of the σ-phase can increase the strength of the alloy, it contributes to a decrease in the ductility of the alloy, which consequently contributes to the obtained shares of index *ERI* and index *PI*. The index *ERI* describes the energy dissipated from a sample after the removal of an indentation load and demonstrates the impact loading resistance of an indented sample [42]. On the other hand, the index *PI* provides information about the mechanical behavior of an indented material during nanoindentation, and it gives the intrinsic plasticity of a material [43]. Analyzing the results for the elastic recovery index, it can be noted that HEA materials dissipated higher energy during nanoindentation measurements, which clearly increases for the alloy with added Mo. At the same time, in addition to stainless steel, it was the reference alloy AlCoCrFeNi that showed the highest plasticity index, which means that it was this alloy (vs. AlCoCrFeNiMo_0.25_) that undergoes almost the maximum plastic deformation during the nanoindentation test. Based on the results of the plasticity index, it can be concluded that the addition of Mo increased the resistance of HEAs to plastic deformation, probably due the presence of the σ phase. The elastic strain to failure is an indicator of the material’s wear resistance, and yield pressure is an indicator of a material’s resistance to plastic deformation [44,45]. The higher the *H*/*E* index, the more dominant elastic deformation is. Alloys with higher values of *H*/*E* tendto show less ductile behavior compared with those featuring lower values [46]. The nanoindentation test suggests that both of these indicators improve after the addition of Mo to the HEA alloy, reaching the highest value. The high wear resistance in the ball-on-disk test for the AlCoCrFeNiMo_0.25_ alloy was confirmed in our earlier work [5].

### 3.3. Corrosion Behavior

Changes in potential (OCP) over time are presented in Figure 4a. An increase in potential towards higher values indicates the tendency of the material to form a protective passive layer on the surface when reaching a state of equilibrium with the system [47]. Among the materials tested, the highest potential value is achieved by AlCoCrFeNiMo_0.25_. In turn, the following behave very similarly: AISI 304L and AlCoCrFeNi.

After the systems reached equilibrium, a non-destructive EIS test was performed. The electrochemical impedance spectrum was simulated using appropriate circuit models to quantitatively determine the resistance and capacity values. Considering the type of electrical equivalent circuit used by Ayyagari et al. [42], and in order to quantitatively determine the polarization resistance values, the EIS data were simulated using a representative circuit shown in Figure 4b, and the determined parameter values were summarized in Table 7. The replacement model consists of a series-connected resistor *R*_s_ representing the resistance of the environment (electrolyte) and a parallel arrangement of *CPE*_1_ and *R*_1_ (i.e., a constant phase element and a resistor representing the transition resistance). The EIS method provides additional information about the electrochemical properties of the passive layer of metal alloys [48,49]. For all tested material groups, the environmental resistance (*R*_s_) changes only slightly, which indicates stable measurement conditions and minimal influence of the solution on the variability of results. The polarization resistance (*R*_1_) is important for the assessment of corrosion resistance. In the case of HEA alloys, higher values of the *R*_1_ parameter were observed compared to austenitic steel. A high polarization resistance (*R*_1_) value indicates good corrosion resistance of the tested materials, with the AlCoCrFeNiMo_0.25_ alloy performing best in this comparison.

When analyzing the results for the parameters of the solid-phase element *CPE*_1_, differences were observed in the value of the coefficient n_1_. Considering the value of the coefficient *n*_1_, it can be seen that all the tested materials behave almost like an ideal capacitor (according to the dependence *n*_1_ → 1). This parameter is related to surface heterogeneity. The results of changes in parameter n1 indicate that the surface of the AlCoCrFeNiMo_0.25_ alloy, which is closest to an ideal capacitor (*n*_1_ = 0.92), is characterized by a compact passive layer free of defects with very good protective properties, while the passive layer of the AlCoCrFeNi alloy appears to be the most heterogeneous (values *n*_1_ = 0.88). Additionally, when analyzing the graphical representations of impedance results in the form of a Bode diagram (Figure 4c), it can be seen that the impedance modulus (|Z|) for all analyzed samples exceeded the value of 10^5^ Ω⋅cm^2^. According to the literature [49,50], fulfilling this relationship for the low-frequency range indicates high corrosion resistance and is characteristic of materials dedicated to medical applications with very high corrosion resistance, such as titanium alloys.

The Bode phase shift diagram (Figure 4d) for the AlCoCrFeNiMo_0.25_ alloy suggests the formation of a stable passive layer on the alloy surface, as the phase angles in the medium- and low-frequency ranges reach their highest values (approximately −80° and −49°, respectively). However, the phase angles for the AlCoCrFeNi alloy drop to approximately −39° at low frequencies, and this behavior indicates that the passive layer forming on the reference HEA alloy contains defects.

The results of the polarization tests in 3.5% NaCl are presented in the form of Tafel curves in Figure 5, and the determined electrochemical parameters are summarized in Table 8. For both HEAs, a favorable shift in the corrosion potential towards higher values relative to stainless steel was observed. However, in the assessment of corrosion resistance, the key parameter is the corrosion current density *i*_corr_, which is also correlated with the corrosion rate. In this respect, the AlCoCrFeNiMo_0.25_ alloy performs best among the materials tested. On the other hand, the HEA alloy without Mo performs less favorably than AISI 304L steel. The conclusion from the polarization tests is that it is the addition of Mo to the HEA material that reduces the corrosion current density and improves corrosion resistance compared to the other materials tested. This effect is explained by Dai et al. [29] as the incorporation of MoO_4_^2−^ oxides into the passive layer dominated mainly by Cr_2_O_3_. The presence of molybdenum oxide in the passive layer of HEA acts as a barrier layer, impeding electrochemical attack. The course of the Tafel curves indicates that all tested materials exhibit a passivation area, after which, in the potential range of 0.12 ÷ 0.26 *V*_SCE_, the anodic current suddenly increases. This behavior results in local breakdown of the passive film, the formation of corrosion pits, and the determination of the pitting potential *E*_pit_ corresponding to the transition of current density from constant to rapid increase. The highest *E*_pit_ value and the most stable and widest passive area are characteristic of AISI 304L stainless steel. The AlCoCrFeNi alloy, in comparison with other materials, showed a gradual and mild increase in current density with observed metastable pitting (at potentials well below *E*_pit_), which underwent repassivation. Meghwal et al. [51] indicate that, apart from *E*_pit_, faster diffusion of Cl^−^ ions occurs on the alloy surface through defects or corrosion-prone phases, which consequently reduces the protective function of the passivation layer, resulting in local corrosion. In general, in HEAs, the addition of Al has a negative effect on the corrosion resistance of alloys. It has been confirmed that the addition of Al significantly reduces the pitting corrosion potential and increases the pitting corrosion area of these alloys in neutral NaCl solutions [52]. Although the HEA alloy had a higher pitting corrosion potential, the passive area was also smaller than that of stainless steel. The fact that AISI 304 stainless steel exhibits a larger passivation area than the HEA alloys compared to it is not unique, as shown in another study [4], where the AlCu_0.5_CoCrFeNiSi alloy had better overall corrosion resistance than stainless steel, but its passivation area was also smaller. As mentioned earlier, the addition of Mo plays a beneficial role in building corrosion resistance. In study [53], the addition of Mo to the Co_1.5_CrFeNi_1.5_Ti_0.5_Mo_x_ alloy proved beneficial, as it significantly increases the pitting potential. Of course, the corrosive environment for HEA alloy applications is also important. Thus, the Co_1.5_CrFeNi_1.5_Ti_0.5_Mo_0.1_ alloy exhibits a wide passivation range in a neutral NaCl corrosive environment and is not susceptible to pitting, but the addition of Mo to the Co–Cr–Fe–Ni–Ti alloy had an adverse effect on corrosion resistance in acidic solutions [53,54]. This effect is related to the formation of a σ phase rich in (Cr, Mo), which in turn reduces the Cr concentration in other phases [53]. However, studies conducted by Shang et al. [31] did not indicate a simple trend in corrosion behavior associated with different Mo_x_ contents (x = 0.1 ÷ 0.5). Improved corrosion resistance was obtained for samples containing Mo0.2 and Mo0.4. The synergistic effect of factors related to the microstructure of HEA alloys and the optimal proportion of Mo and the σ phase was indicated as the reason for the favorable test results.

Figure 6 presents SEM images of the surfaces after HEA corrosion tests, and the results of local EDS analysis are given in Table 9. In the case of the AlCoCrFeNi alloy (Figure 6a), pitting develops inside the grains. Table 4 (spot B) indicates that the grain boundaries are rich in Cr. Therefore, in this case, corrosion occurs due to Cr segregation, where Cr-depleted grain interiors are predisposed to the formation of corrosion pits. Quantitative EDS analysis (Table 9) shows that Al and, to a lesser extent, Ni corrode inside the grains, as a decrease in the concentration of these two elements is mainly observed in comparison with the results obtained before the corrosion tests (see Table 4). Therefore, areas rich in Al and Ni act as the anode, while areas rich in Cr act as the cathode. The reports by Shi et al. [17] also indicate that in AlxCoCrFeNi alloys (x = 0.5 and 0.7), corrosion pits nucleate in areas rich in Al and Ni of the BCC phase, which is more susceptible to Cl¯ ion attack than the FCC phase. In turn, analysis of the mosaic structure of AlCoCrFeNiMo_0.25_ occupied by pits (Figure 6b) indicates that pits develop at the boundaries of areas enriched in Cr and Mo, leading to galvanic corrosion with the adjacent matrix. In this case, EDS analysis (Table 9) clearly indicates a decrease in Al and Ni content in the pitting areas with relatively high Cr and Mo readings. Han et al. [55] point out that Cr-enriched areas representing the σ phase may promote galvanic corrosion at the σ-matrix boundary. On the other hand, areas of material not affected by corrosion pits, occurring in the form of so-called “islands” (Figure 6b, see spot D), may suggest a situation associated with the favorable formation of molybdenum oxide in the passive layer dominated by Cr_2_O_3_, given the locally high Mo and O content.

In order to better visualize the corrosion areas and the role of elements in the corrosion behavior of the tested alloys, Figure 7 presents the EDS analysis in the form of element distribution maps. Pitting corrosion is one of the most common corrosion processes in AISI 304L steel when passive layers are destroyed under the influence of concentrated chloride. The literature data [56,57] indicate that the protective properties of the passive layer of stainless steels are mainly determined by Cr_2_O_3_, but the following iron oxides may also be present: Fe_2_O_3_, FeO, and Fe_3_O_4_. In this case, increased presence of Cr, O, and Fe elements is evident in the immediate vicinity of the pitting. Manganese is generally considered an austenitic element and is added to increase nitrogen solubility [58]. However, its addition is usually accompanied by a reduction in resistance to pitting corrosion, associated with the formation of manganese sulfide (MnS) inclusions, which may constitute pitting initiation sites [3]; small amounts of these were observed as a light blue area on EDS mapping images (Figure 7a). In the case of reference HEA alloys, Cr_2_O_3_ and Fe_2_O_3_ oxides have the largest share in the passive layer, which we associate with the most intense decomposition of these elements at the site of corrosion attack (Figure 7b). The element distribution maps reveal that the areas occupied by Ni and, in particular, Al is subject to corrosion. Wang et al. [59] indicate that Al in passive layers is not stable in the presence of Cl^−^. Therefore, the protection provided by the oxide layer on alloys with a higher Al content is inferior to that on alloys with a lower Al content. Also, Kao et al. [19], investigating the corrosion resistance of Al_x_CoCrFeNi alloys (x = 0, 0.25, 0.5, and 1), found that the protection of the oxide layer on alloys with a higher Al content is worse than on alloys with a lower Al content, because despite the easy formation of a thicker oxide layer, it is porous. The reason for the poorer protective quality of the passive layer is the decrease in the proportion of Cr_2_O_3_ with an increase in Al content. Shi et al. [60] demonstrated that as the Al content in the Al_x_CoCrFeNi alloy increases, the Cr^3+^ and Fe^2+, 3+^ contents decreases. This is because the passive layer containing more Al oxides and hydroxides is thicker but less protective and susceptible to attack by aggressive Cl^−^ ions. It is worth noting that hydroxides in the passive layer have been shown to have a lower density than oxides [61].

Similarly, in the case of the AlCoCrFeNiMo_0.25_ alloy (Figure 7c), a lower presence of Al and Ni is observed in areas where pitting occurs, confirming the susceptibility of these elements to corrosion. In contrast, higher concentrations of O, Cr, and Mo are observed in these areas. Dai et al. [29] have proven that the addition of Mo may slightly thicken the passive layer, improving the protective properties of HEA alloys with Mo in a highly aggressive Cl^−^ environment by modifying the composition and thickness of the passive layer, which is a result of increasing the Cr_2_O_3_/Cr(OH)_3_ ratio and incorporating Mo oxides into the passive layer. With higher amounts of Mo0.2 and Mo0.25 in AlCoCrFeNiMo_x_ alloys, a (Cr, Mo)-rich σ phase appears, acting as a cathode and consequently causing local galvanic corrosion [5]. Galvanic corrosion for CoCrFeNiMo_x_ (x = 0.3–0.5), which occurred at dendrite junctions and at interfaces around the σ phase, was also caused by element segregation with a small anode/cathode interaction area [31]. The obtained oxidation state of Co, Cr, Fe, Ni, and Mo in the passive layer of CoCrFeNiMo_x_ alloys was in the form of Co^2+^, Cr^3+^, Fe^3+^, Ni^2+^, Mo^4+^, and Mo^6+^. Furthermore, it was demonstrated that the oxidation state of elements such as Cr, Fe, and Mo played a leading role compared to their metallic state, while Co and Ni were mainly present in their metallic state. Interestingly, the authors of the study [31] suggested that in the case of Mo, it is Mo^6+^ that plays the main role in delaying corrosion, and not Mo^4+^.

Based on corrosion tests and SEM-EDS analyses of surfaces after corrosion tests in 3.5% NaCl, a diagram of the corrosion mechanism of the tested HEA alloys was prepared (Figure 8). In the case of equiatomic HEA alloy, the (Al, Ni)-rich BCC phase depleted in Cr, which acts as the anode, undergoes local corrosion, which can be compared to the corrosion phenomena occurring in stainless steels (Figure 8a,c). Then, a passive film dominated by Cr_2_O_3_ forms on the cathode surface (at the grain boundaries rich in Cr). As demonstrated by Nie et al. [62], Cr_2_O_33_ is denser and has better protective properties for the matrix than Al_2_O_3_. However, it should be considered that the surface of the alloy has a larger proportion of areas constituting the anode, which translates into a higher proportion of Al_2_O_3_. As a result, the protective properties of the interior of the grains (constituting the matrix) deteriorate as the time spent in the electrolyte solution increases. In turn, in the case of the AlCoCrFeNiMo_0.25_ alloy with a mosaic structure (Figure 8b), there is a potential difference between the phase rich in Cr and Mo precipitates (forming the cathode) and the phase rich in Al and Ni (acting as a microanode—represented in gray). Therefore, areas rich in Al and Ni will be preferentially corroded as a result of the galvanic microcorrosion (Figure 8d). As the lamellar phase constituting the anode undergoes further dissolution, a passive film (rich in Cr_2_O_3_ and MoO_4_) forms between the lamellar areas rich in Cr and Mo. Due to the mosaic structure of the AlCoCrFeNiMo_0.25_ alloy, the anode and cathode areas are balanced, resulting in significantly better corrosion resistance compared to the equiatomic HEA alloy.

## 4. Conclusions

In general, the addition of the element at a level of Mo0.25 significantly affects the microstructure, mechanical properties, and corrosion resistance of the AlCoCrFeNiMo_x_ alloy. Based on the research conducted, it was found that:The addition of Mo creates a mosaic-patterned microstructure, and XRD analysis indicated the presence of the σ phase, whereby the XRD analysis showed that the average grain sizes for B2/BCC phases have a smaller size after Mo addition.Based on Vickers hardness tests, the surface of the HEA alloy with Mo showed a more than 3-fold-increase in hardness compared to AISI 304L, with an increase of nearly 50% compared to the reference HEA, which can be attributed to lattice distortion by Mo, grain refinement, and the presence of a hard and brittle σ phase.Nanoidentification tests have shown that the addition of Mo increases hardness (*H*) and elastic modulus (*E*) and improves the *H/E* and *H*^3^*/E*^2^ ratios, also causing an increase in the elastic recovery index with a decreasing plasticity index. This indicates an improvement in anti-wear properties with resistance to impact loading.Polarization tests in 3.5% NaCl showed that the addition of Mo improves the corrosion resistance of HEA alloys, manifested by a decrease in corrosion current density and, thus, a decrease in corrosion rate and a shift in the corrosion potential towards higher values.EIS corrosion tests showed that the phase angle and impedance modulus values for AlCoCrFeNiMo_0.25_ were higher in the low-frequency range, indicating the formation of a more stable passive layer. Furthermore, in the low-frequency range, all tested surfaces were characterized by high impedance above 10^5^ Ω·cm^2^, indicating that these surfaces have adequate corrosion resistance in a 0.9% NaCl solution, making them suitable for medical applications in this respect. All tested materials behaved almost like an ideal capacitor.SEM-EDS analysis of the surface after the corrosion tests revealed that in the case of the AlCoCrMFeNi equivalent alloy, corrosion occurs due to Cr segregation, where the Cr-depleted grain interiors are predisposed to the formation of corrosion pits, and the Al- and Ni-rich grain interiors are subject to corrosion. Areas rich in Al and Ni form the anode, while areas rich in Cr form the cathode. In the case of HEA with Mo, pits developed at the boundaries of areas enriched in Cr and Mo, leading to galvanic corrosion with an adjacent matrix (rich in Al and Ni).

## Figures and Tables

**Figure 1 materials-18-04566-f001:**
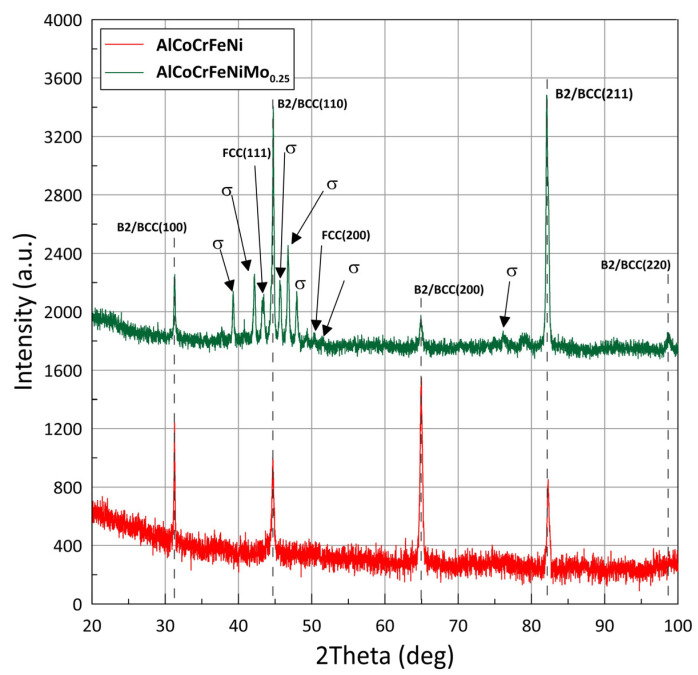
X-ray diffraction patterns of AlCoCrFeNi and AlCoCrFeNiMo_0.25_ high-entropy alloys.

**Figure 2 materials-18-04566-f002:**
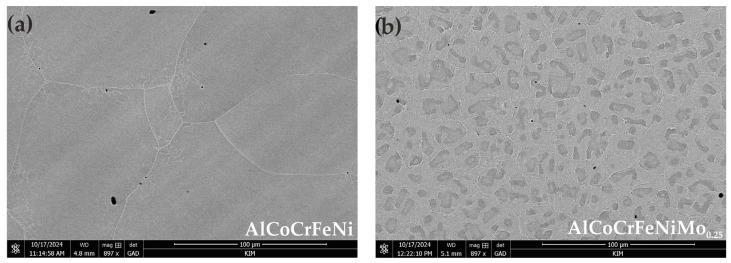
SEM images of the investigated materials with points (A; B and C) of EDS analyses: (**a**,**c**)—AlCoCrFeNi (as-cast); (**b**,**d**,**e**)—AlCoCrFeNiMo_0.25_ (as-cast); and (**f**)—AISI 304L.

**Figure 3 materials-18-04566-f003:**
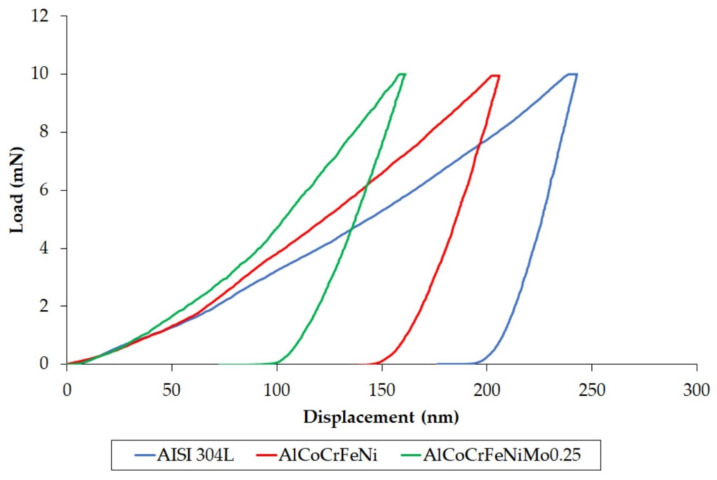
Loading–unloading nanoindentation curves (O&P method) for tested alloys.

**Figure 4 materials-18-04566-f004:**
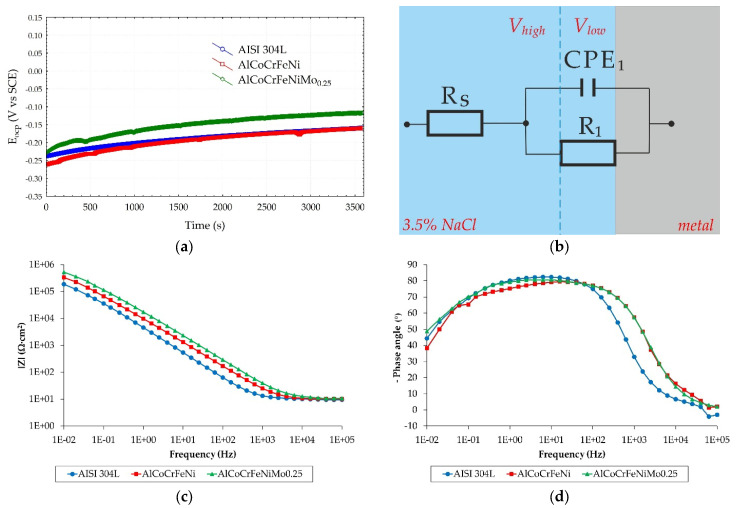
Electrochemical corrosion behavior of tested alloys in 3.5% NaCl solution: (**a**) open-circuit potential variation with time; (**b**) schematic of the circuit used for simulating the EIS; (**c**,**d**) Bode diagrams.

**Figure 5 materials-18-04566-f005:**
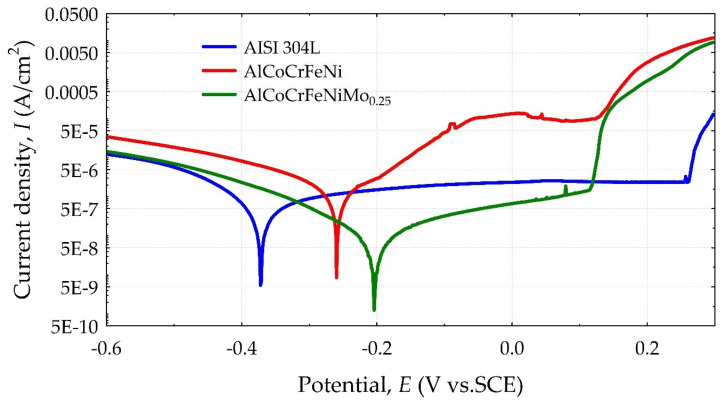
Potentiodynamic polarization curves of tested materials in a 3.5% NaCl solution.

**Figure 6 materials-18-04566-f006:**
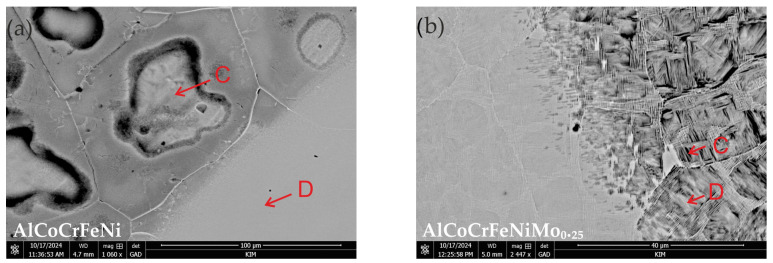
SEM microstructures with points (C and D) of EDS analyses of the surface after corrosion test in 3.5% NaCl solution: (**a**) AlCoCrFeNi; (**b**) AlCoCrFeNiMo_0.25_.

**Figure 7 materials-18-04566-f007:**
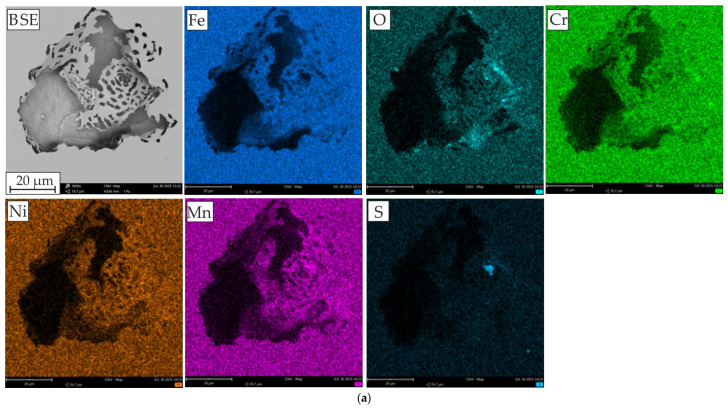
SEM-BSE and EDS mapping images of: (**a**) AISI 304L, (**b**) AlCoCrFeNi, and (**c**) AlCoCrFeNiMo_0.25_.

**Figure 8 materials-18-04566-f008:**
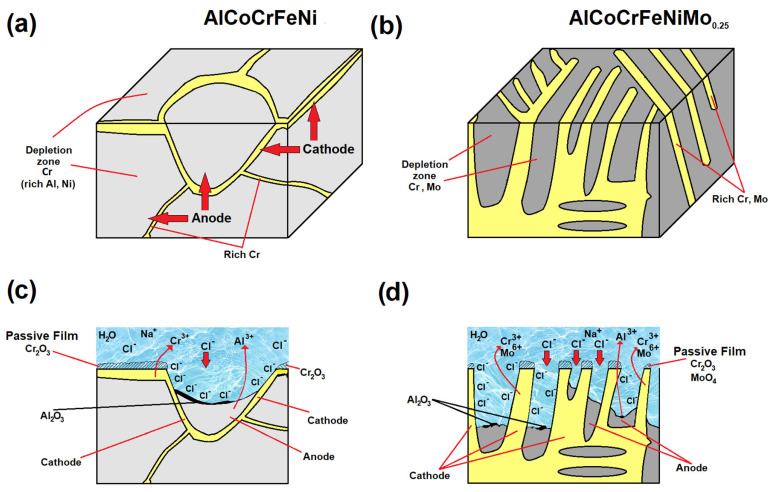
Schematic diagram for corrosion mechanism of AlCoCrFeNi and AlCoCrFeNiMo_0.25_: (**a**,**b**) galvanic corrosion, (**c**) corrosion of the equiatomic HEA, and (**d**) corrosion of the mosaic structure.

**Table 1 materials-18-04566-t001:** Projected and measured chemical composition of investigated HEAs (at. %).

Projected						
Alloy	Al	Co	Cr	Fe	Ni	Mo
AlCoCrFeNi	20.0	20.0	20.0	20.0	20.0	−
AlCoCrFeNiMo_0.25_	19.0	19.0	19.0	19.0	19.0	5.0
**Measured**						
Alloy	Al	Co	Cr	Fe	Ni	Mo
AlCoCrFeNi	18.2	20.2	20.3	20.4	20.8	−
AlCoCrFeNiMo_0.25_	18.1	19.3	18.5	19.0	18.7	5.4

**Table 2 materials-18-04566-t002:** Measured chemical composition of AISI 304L stainless steel (wt.%).

C	Mn	Si	P	S	Cr	Ni	N	Fe
0.028	1.541	0.341	0.045	<0.001	18.23	8.756	0.041	bal.

**Table 3 materials-18-04566-t003:** The grain size and microstrain.

Sample/Phase	Average Grain Size (nm)	Microstrain (%)
AlCoCrFeNi—B2/BCC	90.69	0.096
AlCoCrFeNiMo_0.25_—B2/BCC	73.35	0.057
AlCoCrFeNiMo_0.25_—σ	93.82	0.126

**Table 4 materials-18-04566-t004:** EDS analyses for chemical compositions (at. %) of HEA materials.

Sample	Spot	Phase	Al	Co	Cr	Fe	Ni	Mo
AlCoCrFeNi	A	B2/BCC	20.87	20.17	18.25	19.16	21.55	−
	B	B2/BCC	17.89	19.86	21.57	22.99	22.64	−
AlCoCrFeNiMo_0.25_	A	B2/BCC	19.27	19.18	18.03	18.41	19.0	5.40
	B	B2/BCC	15.13	20.19	20.62	19.83	17.48	6.75
	C	σ	4.06	19.95	34.75	25.41	7.40	8.44

**Table 5 materials-18-04566-t005:** Surface hardness (HV_1_) for different alloys.

Sample	Average Hardness HV_1_	SD (−)
AISI 304L	240	±3.62
AlCoCrFeNi	495	±17.07
AlCoCrFeNiMo_0.25_	772	±27.22

**Table 6 materials-18-04566-t006:** The mechanical properties of tested materials measured in the nanoindentation test.

Material	H (GPa)	E (GPa)	H/E	H^3^/E^2^ (×10^−3^)	W_elast_ (pJ)	W_plast_ (pJ)	W_total_ (pJ)	Index ERI	Index PI
AISI 304L	5.0 ± 0.4	226.4 ± 24.3	0.022	2.37	180.8 ± 11.2	860.4 ± 31.5	1041.7 ± 31.1	0.17	0.83
AlCoCrFeNi	6.2 ± 0.7	194.4 ± 18.8	0.032	6.51	219.1 ± 6.7	652.8 ± 36.2	871 ± 37.0	0.25	0.75
AlCoCrFeNiMo_0.25_	13.3 ± 4.1	261.4 ± 35.9	0.050	37.37	252.2 ± 12.4	361.7 ± 99.1	613.8 ± 100	0.41	0.59

**Table 7 materials-18-04566-t007:** Electrical equivalent circuit parameters of EIS spectra.

Sample No.	R_s_	R_1_	CPE_1_	
			Q_dl1_	n_1_
	Ω·cm^2^	×10^5^ Ω·cm^2^	×10^−5^ Ω^−1^·S^n^·cm^−2^	
AISI 304L	9.95	2.57	3.16	0.90
AlCoCrFeNi	10.49	5.01	1.89	0.88
AlCoCrFeNiMo_0.25_	10.51	6.63	2.09	0.92

*Q*_dl_—constant phase element (*CPE*_1_) of passive film, *n*—exponent of *Q*_dl._

**Table 8 materials-18-04566-t008:** Electrochemical parameters for the tested specimens.

Sample No.	i_corr_	E_corr_	E_pit_	CR
	[µA·cm^−2^]	[mV]	[mV]	×10^−3^ [mm·Year^−1^]
AISI 304L	0.74 ± 0.04	−352 ± 13	255 ± 17	7.75 ± 0.42
AlCoCrFeNi	2.31 ± 0.06	−252 ± 19	159 ± 10	19.68 ± 0.51
AlCoCrFeNiMo_0.25_	0.18 ± 0.03	−202 ± 2	128 ± 22	1.52 ± 0.25

**Table 9 materials-18-04566-t009:** EDS analyses for chemical compositions (at. %) of HEA materials after corrosion test.

Sample	Spot	Al	Co	Cr	Fe	Ni	Mo	O
AlCoCrFeNi	C	16.47	19.36	18.19	19.31	19.66	−	7.00
	D	21.02	23.20	18.28	19.42	19.86	−	1.95
AlCoCrFeNiMo_0.25_	C	0.83	19.54	29.87	22.94	13.07	10.00	3.74
	D	6.04	16.94	22.61	19.34	13.45	12.27	9.35

## Data Availability

The raw data supporting the conclusions of this article will be made available by the authors on request.

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
