# Peer review of "Effect of Adding Molybdenum on Microstructure, Hardness, and Corrosion Resistance of an AlCoCrFeNiMo0.25 High-Entropy Alloy"

_materials, 2025, doi:10.3390/ma18194566_

Round 1
Reviewer 1 Report
Comments and Suggestions for Authors
In this paper, the authors studied the microstructure, phase constitution, and corrosion resistance of AlCoCrFeNi alloy with Mo addition. The results are interesting; however, they are not sufficiently discussed. The following comments should be considered:
1.You studied only one Mo concentration (5 at. % Mo). You should explain why you decided to study this specific composition.
2.The nominal and experimentally measured chemical compositions of the alloys studied are identical (Table 1). Have you observed any difference? What experimental technique was used to determine the chemical concentration of the elements?
3.It would help to specify the powder diffraction file numbers of phases identified in the legend of Fig. 1 (XRD record).
4.Identified phases should be assigned to respective microstructure constituents (Table 4).
5.The microstructure of AISI 304L should be included in Fig. 2 for comparison.
6.Fig. 3 can be transformed into a table to save space in the journal.
7.Corrosion products should be studied using XRD and/or XPS. EDS may not be sufficient to properly identify them.
8.The EDS element maps in Fig. 8 should be properly labeled. The font is extremely small. As such, it is very difficult to identify the individual chemical elements.
9.You should provide a schematic of the corrosion mechanism in the manuscript.
Author Response
Dear Reviewer 1,
We wish to thank the Reviewer for acknowledging our work and recommending it for publication. We also wish to express our gratitude for all comments. We have included the responses to each comment below with the appropriate references from the literature or fragments from the revised article. We would like to mention that all the adjustments made in relation to the comments have been written in red font and highlighted in yellow. We hope that the changes we have made have contributed to the improvement of our manuscript.
The main revisions made in the manuscript and our responses to the Reviewer's comments are as follows:
Reviewer 1: You studied only one Mo concentration (5 at. % Mo). You should explain why you decided to study this specific composition.
Comment 1: To explain why we chose only a specific composition, we must draw attention to one important aspect. All tests were performed on alloys. It is known that the production costs of HEA alloys with this chemical composition are very high. Therefore, efforts were made to further investigate the variant that proved to be the best in earlier tests [doi:10.1007/s43452-025-01260-0]. Based on preliminary mechanical tests (Vickers hardness and sliding wear) and polarization tests, the variant with 5 at. % Mo showed the greatest application potential in terms of the results obtained. At this point, we would like to add that this M0.25 composition is new in terms of chemical composition and has not yet been studied by other scientists. In order to reduce costs, we plan to apply coatings with a composition and characteristics as close as possible to the AlCoCrFeNiMo0.25 alloy variant. We have even conducted initial tests (see photo below) with the application of this type of coating on AISI 304L steel using the following methods: Atmosferic Plasma Spraying (on the left) and laser cladding (on the right).
In the Introduction section, the following text has been added to supplement the justification for undertaking the research:
Based on preliminary mechanical tests (Vickers hardness and sliding wear) and polarization tests in that study, the Mo0.25 variant showed the greatest application potential in terms of the results obtained. Therefore, it was decided to characterize it more extensively in terms of mechanical properties and corrosion resistance. It should be noted that the M0.25 composition is novel in terms of chemical composition and has not yet been studied by other scientists.
Reviewer 1: The nominal and experimentally measured chemical compositions of the alloys studied are identical (Table 1). Have you observed any difference? What experimental technique was used to determine the chemical concentration of the elements?
Comment 2: Thank you for this comment. Indeed there is a mistake. Correct measured chemical composition is as follows:
|
Projected |
||||||
|
Alloy |
Al |
Co |
Cr |
Fe |
Ni |
Mo |
|
AlCoCrFeNi |
20.0 |
20.0 |
20.0 |
20.0 |
20.0 |
− |
|
AlCoCrFeNiMo0.25 |
19.0 |
19.0 |
19.0 |
19.0 |
19.0 |
5.0 |
|
Measured |
||||||
|
Alloy |
Al |
Co |
Cr |
Fe |
Ni |
Mo |
|
AlCoCrFeNi |
18.2 |
20.2 |
20.3 |
20.4 |
20.8 |
− |
|
AlCoCrFeNiMo0.25 |
18.1 |
19.3 |
18.5 |
19.0 |
18.7 |
5.4 |
Part of Table 1 dealing with measured chemical composition was corrected. Chemical composition of studied alloys was determined using SEM-EDS technique from the large area.
Reviewer 1: It would help to specify the powder diffraction file numbers of phases identified in the legend of Fig. 1 (XRD record).
Comment 3: The ICDD database (PDF-4+ program) was not used during phase analysis. High Score Plus sofware (Panalytical) with Crystallography Open Database files was used for phase identification. The corresponding CIF file numbers used for the simulations are provided in the text describing the XRD profiles and individual phases. The file for NiAl (CsCl-type), COD no. 9008802, was used to identify the B2/BCC phase; COD no. 7204808 for the FCC phase, and COD no. 2106167 for the sigma phase.
The above comment has been taken into account in the work, and the description after the changes is as follows:
X-ray diffraction patterns of AlCoCrFeNi and AlCoCrFeNiMo0.25 high entropy alloy samples are shown in Figure 1. XRD patterns of AlCoCrFeNi showed the diffraction peaks located at around 2θ = 31.25°, 44.67°, 64.91° and 82.25°. As a result of analysis using High Score Plus software package with Crystallography Open Database (COD), it was found that for the peak reveals in XRD profiles, the peaks positions correspond to phases B2/BCC (COD no. 9008802) with the cubic structure and the space group Pm-3m. The occurrence of a peak at 2θ = 31.25° and the absence of peaks around 2θ = 54.8° and 72.9° indicate incomplete phase ordering. However, XRD patterns of AlCoCrFeNiMo0.25 showed diffraction peaks confirming B2/BCC phases located at around 2θ = 31.25°, 44.67°, 64.91° and 82.25°, and new relatively low-intensity peaks at 43.23° and 50.38° representing the FCC with the cubic structure and the space group F/m-3m (COD no. 7204808). Additionally, XRD profile reveals peaks at 2θ = 42.16°, 44.61°, 45.67°, 47.93° and 49.29° and 51.39°, representing σ-phase with a tetragonal crystal structure and the space group P/42mnm (COD no. 2106167). The above results indicate that adding small amount of Mo atoms to significantly change the structure of AlCoCrFeNi high entropy alloy. The individual phases are marked at the diffraction peaks in Figure 1.
Reviewer 1: Identified phases should be assigned to respective microstructure constituents (Table 4).
Comment 4:We have taken the reviewer's comment into Table 4.
Reviewer 1: The microstructure of AISI 304L should be included in Fig. 2 for comparison.
Comment 5: We have taken the reviewer's comment into account and included the microstructure of AISI 304L steel (Figure 2f).
Figures 2f show the equiaxed coarse microstructure of AISI 304L stainless steel (after electrolytic etching) consisting of austenite with visible twins. The crystal structure of the austenite phase is FCC.
Reviewer 1: Fig. 3 can be transformed into a table to save space in the journal.
Comment 6: We have taken the reviewer's comment into account and transformed Fig. 3 into Table 5.
Table 5. Surface hardness (HV1) for different alloys.
|
Sample |
Average hardness HV1 |
SD (−) |
|
AISI 304L |
240 |
±3.62 |
|
AlCoCrFeNi |
495 |
±17.07 |
|
AlCoCrFeNiMo0.25 |
772 |
±27.22 |
Reviewer 1: Corrosion products should be studied using XRD and/or XPS. EDS may not be sufficient to properly identify them.
Comment 7: EDS analysis is very often used by researchers as a basic tool for describing the surface of materials after corrosion tests. Unfortunately, we did not have access to XPS, and the spectrum obtained from XRD was too weak to indicate specific oxides. Nevertheless, based on our SEM-EDS analysis, we can confidently identify the elements and areas that are subject to corrosion. We hope that the graphical presentation of the materials after corrosion testing, supported by qualitative and quantitative SEM-EDS analysis, combined with references to literature studies with XPS analysis (in section 3.3) and the proposed corrosion scheme, will compensate for this shortcoming. In our next analysis of corrosion tests planned for HEA coatings, we will certainly attempt to include this XPS analysis.
Reviewer 1: The EDS element maps in Fig. 8 should be properly labeled. The font is extremely small. As such, it is very difficult to identify the individual chemical elements.
Comment 8: We have taken the reviewer's comment into account and included a new version of Fig. 7 (currently). We hope that the revised edition of the figure is much clearer.
(a)
(b)
(c)
Reviewer 1: You should provide a schematic of the corrosion mechanism in the manuscript.
Comment 9: We have included the diagram in the manuscript as requested by the reviewer.
Figure 8. Schematic diagram for corrosion mechanism of AlCoCrFeNi and AlCoCrFeNiMo0.25: a) and b) galvanic corrosion, c) corrosion of equiatomic HEA, d) corrosion of the mosaic structure
Based on corrosion tests and SEM-EDS analyses of surfaces after corrosion tests in 3.5% NaCl, a diagram of the corrosion mechanism of the tested HEA alloys was prepared (Figure 8). In the case of equiatomic HEA alloy, the (Al, Ni)-rich BCC phase depleted in Cr, which acts as the anode, undergoes local corrosion, which can be compared to the corrosion phenomena occurring in stainless steels (Figure 8a,c). Then, a passive film dominated by Cr2O3 forms on the cathode surface (at the grain boundaries rich in Cr). As demonstrated by Nie et al. [62] Cr2O3₃ is denser and has better protective properties for the matrix than Al2O3. However, it should be considered that the surface of the alloy has a larger proportion of areas constituting the anode, which translates into a higher proportion of Al2O3. As a result, the protective properties of the interior of the grains (constituting the matrix) deteriorate as the time spent in the electrolyte solution increases. In turn, in the case of the AlCoCrFeNiMo0.25 alloy with a mosaic structure (Figure 8b), there is a potential difference between the phase rich in Cr and Mo precipitates (forming the cathode) and the phase rich in Al and Ni (acting as a microanode - represented in grey). Therefore, areas rich in Al and Ni will be preferentially corroded as a result of the galvanic microcorrosion (Figure 8d). As the lamellar phase constituting the anode undergoes further dissolution, a passive film (rich in Cr2O3 and MoO4) forms between the lamellar areas rich in Cr and Mo. Due to the mosaic structure of the AlCoCrFeNiMo0.25 alloy, the anode and cathode areas are balanced, resulting in significantly better corrosion resistance compared to the equiatomic HEA alloy.

Reviewer 2 Report
Comments and Suggestions for Authors
This manuscript reported the improved corrosion resistance of AlCoCrFeNi alloy with Mo addition. It is fairly well written. I recommend acceptance with minor revision.
-Abstract: Please rewrite the abstract. Abstract should be concise and you should only mention the major findings of this study. It is not for details of experimental conditions.
-Fig8. Please indicate the element on the EDS maps. The scale annotation is too small to see.
Author Response
Dear Reviewer 2,
We wish to thank the Reviewer for acknowledging our work and recommending it for publication. We also wish to express our gratitude for all comments. We have included the responses to each comment below with the appropriate references from the literature or fragments from the revised article. We would like to mention that all the adjustments made in relation to the comments have been highlighted in yellow. We hope that the changes we have made have contributed to the improvement of our manuscript.
The main revisions made in the manuscript and our responses to the Reviewer's comments are as follows:
Reviewer 2: -Abstract: Please rewrite the abstract. Abstract should be concise and you should only mention the major findings of this study. It is not for details of experimental conditions.
Comment 1: We have taken the reviewer's suggestions into account, it was decided to modify the abstract, which now reads as follows:
Recent literature reports have shown that individual HEAs have especially those of the AlCoCrFeNi composition system, alloyed with appropriately selected elements, excellent mechanical properties and corrosion resistance, making them promising candidates for replacing conventional materials such as austenitic steels in corrosive environments. Therefore, in the present study, the high entropy alloy AlCoCrFeNiMo0.25 was examined and compared with AISI 304L steel and the reference alloy AlCoCrFeNi. HEA was produced by arc melting in vacuum. The effect of molybdenum addition (5% at.) on the structure, mechanical properties and corrosion resistance was evaluated. Potentiodynamic polarization and electrochemical impedance spectroscopy tests were carried out in a 3.5% NaCl solution in a three-electrode electrochemical system. The addition of molybdenum to AlCoCrFeNiMox alloy additionally causes, along with the BCC phase, the formation of σ phase and FCC phase (less than 1%), as well as changes in the microstructure, leading to the fragmentation of grains and obtaining a mosaic structure. On the basis of nanoindentation tests, it was established that the addition of Mo increases hardness and elastic modulus, and improves nanoindentation coefficients H/E and H3/E2, as well as an increase in elastic recovery index with decreasing plasticity index (vs. reference equiatomic HEA). That indicates the improvement of anti-wear properties with impact loading resistance. In turn, electrochemical tests have shown that the addition of Mo improves corrosion resistance. Corrosion pitting develops in Al- and Ni-rich areas of HEA alloys, as a result of the galvanic microcorrosion which is related to Cr chemical segregation. In general, the addition of 5% Mo results in a fine-grained mosaic structure, which primarily translates into favorable nanoindentation and corrosion properties of the AlCoCrFeNiMo0.25 alloy.
Reviewer 2: -Fig8. Please indicate the element on the EDS maps. The scale annotation is too small to see.
Comment 2: A similar comment was made by Reviewer No. 1 in point 8. The above comment has been taken into account in the paper. Currently, after the changes, it is Fig.7

Reviewer 3 Report
Comments and Suggestions for Authors
The manuscript has many discrepancies.
- The introduction is somewhat lengthy, but the research gap is not sharply defined.
- Comparison with 304L doesn't make any sense, because properties like hardness are orders of magnitude different due to fundamentally different microstructures.
- The deformation mechanism was not properly explained. 5 s of holding time during nanoindentation doesn't make any sense for creep. Atleast, 30 s to 1 min. is required for creep study.
- What is the role of sigma phase on mechanical behavior of the Mo containing HEA? What is the grain size distribution of the sigma phase, and how it is different from the matrix grain size. Also elemental distribution (EDX mapping) of the matrix and sigma phase is required for clear difference. High magnification image showing the sigma phase and the matrix is mandatory.
- The discussion of strengthening and deformation mechanisms in the manuscript is somewhat limited. While hardness and nanoindentation indices (H/E, H³/E², ERI, PI) are reported, the underlying deformation mechanisms are not fully explored. I encourage the authors to consider relevant recent literature that addresses strain-hardening mechanisms in multi-principal element alloys. For example, the paper “Enhanced Strain Hardening Response of Multi-Principal Element Alloys with L1â‚‚ Nanodomains Designed for Multiple Objectives and Constraints” could provide useful context for interpreting how nanoscale structural features or secondary phases (e.g., σ phase in your system) influence strain hardening, strength–ductility balance, and overall mechanical response.
- Surprisingly, there is no heat treatment process involved after arc melting. There should be solutionizing heat treatment at certain temperature to homogenize the distribution of the elements. The authors need to comment on this.
Author Response
Dear Reviewer 3,
We wish to thank the Reviewer for acknowledging our work and recommending it for publication. We also wish to express our gratitude for all comments. We have included the responses to each comment below with the appropriate references from the literature or fragments from the revised article. We would like to mention that all the adjustments made in relation to the comments have been highlighted in yellow. We hope that the changes we have made have contributed to the improvement of our manuscript.
The main revisions made in the manuscript and our responses to the Reviewer's comments are as follows:
Reviewer 3: The introduction is somewhat lengthy, but the research gap is not sharply defined.
Comment 1: All tests were performed on alloys. It is known that the production costs of HEA alloys with this chemical composition are very high. Therefore, efforts were made to further investigate the variant that proved to be the best in earlier tests [doi:10.1007/s43452-025-01260-0]. Based on preliminary mechanical tests (Vickers hardness and sliding wear) and polarization tests, the variant with 5 at. % Mo showed the greatest application potential in terms of the results obtained. At this point, we would like to add that this M0.25 composition is new in terms of chemical composition and has not yet been studied by other scientists. In order to reduce costs, we plan to apply coatings with a composition and characteristics as close as possible to the AlCoCrFeNiMo0.25 alloy variant. We have even conducted initial tests (see photo below) with the application of this type of coating on AISI 304L steel using the following methods: Atmosferic Plasma Spraying (on the left) and laser cladding (on the right).
In the Introduction section, the following text has been added to supplement the justification for undertaking the research:
Based on preliminary mechanical tests (Vickers hardness and sliding wear) and polarization tests in that study, the Mo0.25 variant showed the greatest application potential in terms of the results obtained. Therefore, it was decided to characterize it more extensively in terms of mechanical properties and corrosion resistance. It should be noted that the M0.25 composition is novel in terms of chemical composition and has not yet been studied by other scientists.
Reviewer 3: Comparison with 304L doesn't make any sense, because properties like hardness are orders of magnitude different due to fundamentally different microstructures.
Comment 2: Unfortunately, we cannot fully agree with the above claim. The concept of the article is also related to the search for alternative materials to stainless steels, which have an established position in terms of applications requiring, above all, high corrosion resistance, as emphasised in the Introduction. It is well known that steel, despite its relatively favourable corrosion parameters, has unsatisfactory mechanical properties. There are a number of application areas where, in addition to high corrosion resistance, other mechanical properties are required. Therefore, in dozens of publications devoted to the corrosion resistance of HEA alloys, austenitic steels are used as a reference material (most often, steel grades from the 300 series are chosen as representatives). At the same time, there are a whole host of articles related to the characteristics of HEA, where mechanical properties (including hardness, strength, wear resistance, etc.) are compared with stainless steels
(https://doi.org/10.1080/21663831.2014.912690 , https://doi.org/10.1016/j.wear.2019.203028 , https://doi.org/10.1016/j.ijoes.2023.100074, https://doi.org/10.1016/j.actamat.2023.119280 ) depending on the context of the application. We agree with the reviewer that in terms of hardness and structural composition, stainless steels are diametrically different materials. Bearing in mind, above all, the reliable characteristics of the materials under study, we wanted to take a comprehensive approach to the presentation of the results in order to avoid the accusation that, since we are testing three materials, it is also worth referring to them in the same way in all aspects of the study. At this point, we would like to mention that although conventional AISI 304L steel is such a well-known material, we could not avoid the suggestion from Reviewer No. 1 (see point 5) to add the microstructure of AISI 304L steel to the microstructures in Fig. 2. Therefore, we believe that our approach to presenting the results not only from corrosion but also from nanoindentation and Vickers hardness tests for all materials is correct.
Reviewer 3: The deformation mechanism was not properly explained. 5 s of holding time during nanoindentation doesn't make any sense for creep. At least, 30 s to 1 min. is required for creep study.
Comment 3: We agree with the reviewer that the holding time of 5 seconds during creep determination may be too short. In response to this objection, we noticed that a mistake had been made, as the holding time was 10 seconds, and in order to avoid being accused of making arbitrary changes, we are including a scan of the test report from the device. Nevertheless, nowhere in the paper were the CIT values characterizing creep deliberately given. The authors of the paper wanted to emphasise the differences in the characteristics of the O&P curves of the tested materials at Fmax. In fact, the word ‘creep’ may have sounded a little exaggerated in this context. Therefore, this part of the description of the O&P curves has been rewritten. Regarding the choice of holding time, norm ISO 14577-1:2015 does not directly indicate the value of time selection. There is also the following reference – „A 60 s hold period to measure thermal drift can also be required (see Annex G)”. However, in Annex G, the standard refers us to a literature reference stating – „Reference [15] proposes hold periods dependent on the material type range from 8 s for fused quartz to 187 s for aluminium”. However, in the literature to which the standard refers, i.e. Surface and Coatings Technology 148 (2001) 191–198 we find Table 2 with “Recommended hold periods for the investigated materials” - and for steel, for example, it is 12 seconds. In addition, there are studies (see DOI: 10.1007/s11665-016-2082-8 – 31 citations and DOI: 10.1007/s11665-015-1576-0 – 57 citations ) for HEA alloys with the composition Al-Co-Cr-Fe-Ni, where the indenter was held for 10 s during the holding stage to determine whether a creep behavior occurred.
At this point, I would like to note that we are aware that for many conventional materials that analyse the phenomenon of creep in detail based on nanoindentation methods, the holding time is significantly longer. However, it should be noted that HEA alloys are still a new group of materials that are not fully understood, and it is difficult to ignore even these two works, which nevertheless enjoy high citation rates. Therefore, we hope that the above arguments, the premises that guided the authors, and the changes in the manuscript text will be accepted.
The following amendments have been included in section 3.2;
As can be observed from the curves, at 10-second intervals, all tested materials may exhibit a tendency to creep, as the displacement of the diamond indenter increased at a constant maximum load of Fmax = 10 mN. Although creep changes are analysed at much higher holding time value [39], for HEA alloys based on the Al-Co-Cr-Fe-Ni composition, the authors of the papers [40] and [41] used a holding time of 10 s to determine whether creep occurred. Jiao et al. [41] demonstrated that for an equimolar alloy, a significant accumulation of material around the indenter suggests that strongly localised plastic deformation occurs. In the case of this type of HEA alloy, the deformation mechanism during nanoindentation is explained by an increase in dislocation density, which increases as the indentation depth decreases. The dislocations then spread to smaller slip circles, which can lead to nanoindentation creep.
Screenshot from the nanoindentation test report with information used in the tests.
Reviewer 3: What is the role of sigma phase on mechanical behavior of the Mo containing HEA? What is the grain size distribution of the sigma phase, and how it is different from the matrix grain size. Also elemental distribution (EDX mapping) of the matrix and sigma phase is required for clear difference. High magnification image showing the sigma phase and the matrix is mandatory.
Comment 4: The grain sizes are listed in Table 3. In turn, information on the role of the sigma phase and the strengthening resulting from the addition of Mo to HEA alloys was supplemented with the following explanation included in the manuscript text:
As illustrated in Figure 2e, the structure identified as sigma phase separation is displayed. The chemical composition obtained for this phase in the EDS analysis (Table 4, spot C) is similar to the results obtained by Zhu et al. [28] for the sigma phase occurring in the Mo0.2 alloy variant. In contrast to the theoretical thermodynamic information determined using the ThermoCalc program presented in [36], in addition to the typical elements for this phase (i.e., Cr, Fe, Ni, and Mo), EDS analysis (which has its technical limitations) even at high magnifications indicates elements from the matrix such as Co and small amounts of Al. In the present case, the interlayer distance of the phase was found to be in the range of 100–300 nm. It is important to note that Zhu et al. [28] observed that the interlayer distances of phase decrease with increasing Mo content in the alloy.
And
σ-phase is unintentionally obtained. As the content increases, the proportion of the BCC phase decreases at the expense of the σ phase. Consequently, the total hardness of the HEA alloy increases, as the hardness of the σ phase is significantly higher than that of the BCC phase. As mentioned earlier, the σ phase is a very hard and brittle intermetallic phase. It increases the hardness of the alloy but reduces its ductility, as has already been demonstrated in studies [28] [35].
In our previous studies [5] it has been demonstrated that the hardness of the alloy also increases with the increase in Mo content, which entails a restructuring of the coarse-grained structure to obtain a fine mosaic-patterned structure. In addition to the presence of the sigma phase, it is primarily the formation of a plate-like structure that contributes to the increase in the phase boundary, which in turn translates into an increase in the strength of HEA alloys with Mo [5] [28].As illustrated (Figure 2 and Table 3), Mo is present in both light and dark plates, with an increased Mo content observed in lighter plates. The sigma phase occurs in plates with an increased Mo content, i.e. light plates. As Zhu et al. [28] indicated, the strength of HEA material improves significantly with increasing Mo content, while the ductility of the alloy is weakened. This is due to the fact that the ductility of the sigma phase is lower than that of the BCC phase. Mo occurs in the disordered BCC phase. As a reminder, the atomic radius of Mo is 190 pm, and adding Mo as an element with a larger atomic size can increase the lattice constant of the disordered BCC phase. As a result, its value approaches the lattice constant of the ordered B2 phase. Furthermore, Zhu et al. [28] indicate that the increase in hardness and decrease in ductility of HEA alloys with Mo can be attributed to the fact that the number of slip systems in a complex structure (BCC+σ) is lower than that of a BCC structure. This also explains why the ductility of the σ phase is significantly lower than that of BCC.
Reviewer 3: The discussion of strengthening and deformation mechanisms in the manuscript is somewhat limited. While hardness and nanoindentation indices (H/E, H³/E², ERI, PI) are reported, the underlying deformation mechanisms are not fully explored. I encourage the authors to consider relevant recent literature that addresses strain-hardening mechanisms in multi-principal element alloys. For example, the paper “Enhanced Strain Hardening Response of Multi-Principal Element Alloys with L1â‚‚ Nanodomains Designed for Multiple Objectives and Constraints” could provide useful context for interpreting how nanoscale structural features or secondary phases (e.g., σ phase in your system) influence strain hardening, strength–ductility balance, and overall mechanical response.
Comment 5: Thank you for your suggestion. We have reviewed the work you mentioned and added the following passage to the manuscript:
Grain refinement in the case of the mosaic structure of the AlCoCrFeNiMo0.25 alloy may determine the Hall-Petch effect, contributing, among other things, to the strengthening of the material and increasing its strength [5]. As pointed out by Xu et al. [37] secondary phases influence strain hardening, strength–ductility balance, and overall mechanical response. Xi and Li [38] indicate that the presence of the σ phase leads to increased enhances alloy strength by anchoring grain boundaries and impeding their movement. This, in turn, prevents the formation of twinning, which reduces the ductility. The increase in material strength can also be explained by the effect of σ phase on the grain boundaries blocking sliding motion and dislocation motion.
Reviewer 3: Surprisingly, there is no heat treatment process involved after arc melting. There should be solutionizing heat treatment at certain temperature to homogenize the distribution of the elements. The authors need to comment on this.
Comment 6: It is widely known that these type of the alloys exhibit irreversible phase change at temperature about 630℃. Moreover, in the frame of other work Heat treatment has been done. As the effect formation of round precipitates was observe and resulted in severe segregation of the elements. To avoid these phenomena our work was focused on studies on HEA in the as-cast conditions. Such information was introduced into the text. No modification at this point was done.
|
a) AlCoCrFeNiMo0.25 as-cast |
b) AlCoCrFeNiMo0.25 heat-treated |
|
|
|
* unpublished results. Please do not used in any form.

Round 2
Reviewer 1 Report
Comments and Suggestions for Authors
Authors answered my comments. The paper can be accepted for publication.
Reviewer 3 Report
Comments and Suggestions for Authors
Good for publication.